# *HSPB6* Is Depleted in Colon Cancer Patients and Its Expression Is Induced by 5-aza-2′-Deoxycytidine In Vitro

**DOI:** 10.3390/medicina59050996

**Published:** 2023-05-21

**Authors:** Bader O. Almutairi, Mikhlid H. Almutairi, Abdulwahed F. Alrefaei, Saad Alkahtani, Saud Alarifi

**Affiliations:** Department of Zoology, College of Science, King Saud University, P.O. Box 2455, Riyadh 11451, Saudi Arabia; malmutari@ksu.edu.sa (M.H.A.); afrefaei@ksu.edu.sa (A.F.A.); salkahtani@ksu.edu.sa (S.A.); salarifi@ksu.edu.sa (S.A.)

**Keywords:** HSPB6, colon cancer, DNA methylation, cell lines, gene expression

## Abstract

*Background and Objectives:* Colon cancer (CC) is the second most common cancer in Saudi Arabia, and the number of new cases is expected to increase by 40% by 2040. Sixty percent of patients with CC are diagnosed in the late stage, causing a reduced survival rate. Thus, identifying a new biomarker could contribute to diagnosing CC in the early stages, leading to delivering better therapy and increasing the survival rate. *Materials and Methods: HSPB6* expression was investigated in extracted RNA taken from 10 patients with CC and their adjacent normal tissues, as well as in DMH-induced CC and a colon treated with saline taken from a male Wistar rat. Additionally, the DNA of the LoVo and Caco-2 cell lines was collected, and bisulfite was converted to measure the DNA methylation level. This was followed by applying 5-aza-2′-deoxycytidine (AZA) to the LoVo and Caco-2 cell lines for 72 h to see the effect of DNA methylation on *HSPB6* expression. Finally, the GeneMANIA database was used to find the interacted genes at transcriptional and translational levels with *HSPB6*. *Results:* We found that the expression of *HSPB6* was downregulated in 10 CC tissues compared to their adjacent normal colon tissues, as well as in the in vivo study, where its expression was lower in the colon treated with the DMH agent compared to the colon treated with saline. This suggests the possible role of *HSPB6* in tumor progression. Moreover, *HSPB6* was methylated in two CC cell lines (LoVo and Caco-2), and demethylation with AZA elevated its expression, implying a mechanistic association between DNA methylation and *HSPB6* expression. *Conclusions*: Our findings indicate that *HSPB6* is adversely expressed with tumor progression, and its expression may be controlled by DNA methylation. Thus, *HSPB6* could be a good biomarker employed in the CC diagnostic process.

## 1. Introduction

Colon cancer (CC) is diagnosed in both genders, and a published report by the Global Cancer Observatory shows that CC represented 6.2% of diagnosed cancers and 5.8% of deaths in 2020 [1,2]. In Saudi Arabia, CC ranks second in cancer incidence, and less than a third of patients are diagnosed with localized tumors [3,4]. Additionally, the number of new CC cases has increased by 8% in the last decade among men and could continue to increase by 60% by 2040 [5]. However, in developed countries, due to screening programs and better treatment, CC cases and mortality rates have been reduced [6]. Subsequently, there has been a demand to develop a screening program to detect early-stage CC, which may contribute to increasing the survival rate and delivering better treatment [4].

CC is formed in the epithelium cells located in the colon, which harbor differentiated cells, including goblet cells, enteroendocrine cells, and enterocytes [7]. These cells are normally replaced every week to form a new epithelium with multipotent stem cells [7]. Therefore, any dysregulation during the process from proliferation to differentiation, such as an abnormality of gene expression or epigenetic modifications, may result in the production of uncontrolled dividing cells, leading to tumor formation [8]. Despite the ambiguity of the pathological pathways of CC initiation, some abnormal genes are implicated in accelerating the growth of CC and are involved in chemoresistance, including *CXCL12*, *SUL1A1* [9], *ADAM2*, *SPZ1* [10], *CTAG1A*, and *TKTL2* [11]. Additionally, epigenetic events, particularly DNA methylation, contribute to CC tumorigenesis, either by silencing tumor suppressor genes *SEPT9* [12] or *FOXO1* [13] or activating oncogenes *MAGE*-B1 [14], *MYBL2*, and *AXIN2* [15]. Thus, investigating the alternation of gene expression and epigenetic events during tumorigenesis could provide promising targets that can be utilized as predictive, prognostic, and diagnostic biomarkers [16].

Heat shock proteins (HSPs) are present in the mammalian heart, providing cardioprotection. These proteins are divided into two main groups based on their molecular size: the large proteins include HSP40, HSP60, HSP70, HSP90, and HSP110 [17]. In contrast, the small proteins contain HSP20 (HSPB6), HSP22, and HSP27 [17]. Although HSP expression is found in normal and cancer cells, it could play a role in intercellular cross-talk and increase the survival rate of cancer cells through activating anti-apoptotic genes as well as chemoresistance [18]. *HSPB6* is located on small HSPs and has several functions, including molecular chaperoning [19]. *HSPB6* expression has been found to be depleted in ovarian cancer [20] and colorectal cancer [21], and its presence drives HCT116 cells to undergo apoptosis events by inducing caspase 3/7 expression and inhibiting BCL2 expression [21]. Its absence has also been detected in lung cancer, suggesting its role in cancer prevention [22]. Additionally, a report on tumor and non-tumor tissues extracted from patients with hepatocellular carcinoma showed that the expression of *HSPB6* was elevated in the non-tumor tissues compared to the tumor tissues [23], and it has a direct link with *BAX* to regulate apoptosis [24]. Although *HSPB6* has the function of decreasing cell immigration [25], it has also been associated with poor outcomes in bladder cancer [26] and contributes to cancer growth in esophageal cancer [27]. Consequently, an abnormal expression of *HSPB6* is arguably involved in tumorigenesis [28].

Accumulating evidence has indicated that changes in *HSPB6* DNA methylation levels during tumor formation have been observed [29], affecting its gene expression [22], as well as having an aberrant expression in several cancers, including CC [21]. Thus, we investigated the expression of *HSPB6* in CC tissues and their adjacent normal tissues (NC) collected from patients with CC among the Saudi population. We also unveiled the effect of DNA methylation on its expression.

## 2. Materials and Methods

### 2.1. Bioinformatic Analysis

R2 genomic analysis and visualization platform (https://hgserver1.amc.nl/cgi-bin/r2/main.cgi, accessed on 2 September 2022 ) is an online platform containing plenty of publicly available genomic data. Thus, this platform was utilized to compare the expression of *HSPB6* in (GSE8671); it contains 32 samples extracted from patients with CC and their adjacent NC. Additionally, the Kaplan–Meier scanner was employed to associate the survival rate and *HSPB6* expression in 466 samples derived from microarray analysis of mixed colon adenocarcinoma (GENCODE36) at a cutoff value of 1.33. The GeneMANIA database (genemania.org, accessed on 15 October 2022 ) is a website used to predict and generate hypotheses about gene networks. Thus, it was employed to produce physically and genetically connected genes with *HSPB6* [30].

### 2.2. Ethical Approval

We obtained two ethical approvals to collect samples from patients with CC and induce CC formation in male Wistar rats from the Al-Imam Muhammad Ibn Saud Islamic University (HAPO-01-R-011, July 9, 2020) [14] and the King Saud University Ethics of Scientific Research Committee (KSU-SE-20-20. May 8, 2020) [30], respectively. Ten patients with CC signed a consent form and provided us with their ages, personal medical information, and social behavior [14]. The 10 male Wistar rats were provided by the animal house at the College of Science, King Saud University [30].

### 2.3. Sample Collections

Ten CC tissues and their adjacent NC tissues were collected from patients with CC who had not been subjected to chemotherapy or radiotherapy. The NC tissues were collected at a minimum distance of 5 cm from the region containing the CC tissue samples. They were confirmed at King Khalid University Hospital by a team of surgeons and pathologists for their eligibility to enroll in this study. Colon tissues from male Wistar rats were also taken from the normal control group and the DMH-induced group [31]. All tissues were immediately placed in Eppendorf tubes containing RNAlater [14,31].

### 2.4. Induction of Colon Cancer in Wistar Rats

Before we commenced the course of cariogenic reagent (DMH) treatment, 10 male Wistar rats (weighing 120–150 g) were selected from the animal house at the College of Science, King Saud University [32] and accommodated in polyethylene cages under standard laboratory conditions for 7 days for adaptation. The DMH (Sigma-Aldrich) was obtained and dissolved in 1 mM EDTA, followed by the addition of 1 M sodium hydroxide to reach pH 6.5 [32]. The animals were divided into two groups: the control group with five Wistar rats treated with saline via subcutaneous injection for 5 weeks and the DMH-induced group with five Wistar rats treated with 40 mg/kg b.wt of DMH twice a week for 5 weeks via subcutaneous injection [32].

### 2.5. Cell Culture and 5-Aza-2′-Deoxycytidine Induction

The CC cell lines (LoVo and Caco-2) were obtained from the American Type Culture Collection (ATCC, USA). The cells were grown in DMEM media (Sigma-Aldrich) supplied with 10% FBS and 10,000 U/mL antibiotics and left at 37 °C in a 5% CO_2_ incubator. After three passages, cells were seeded in T25 flask at density of 8 × 10^5^ and incubated with 1 μM 5-aza-2′-deoxycytidine (Sigma) for 72 h, refreshing media every 24 h. The control cultures received the same volumes of DMSO (drug solvent) [33].

### 2.6. Genomic DNA Extraction, Bisulfite Treatment, and Methylation-Specific PCR (MSP)

The LoVo and Caco-2 cells were collected from the flasks and pelleted before a GeneJET genomic DNA purification kit (Thermo Scientific) was applied. This was followed by measuring the concentration and purity of DNA via the Nano-Drop8000 spectrophotometer (Thermo Fisher Scientific). A total of 500 ng DNA was then treated with bisulfite conversion reagent EZ DNA Methylation-Gold Kit (Zymo Research) [34]. Two sets of *HSPB6* MSP primers were designed using (http://www.urogene.org/cgi-bin/methprimer/methprimer.cgi, accessed on 15 September 2022) [35]. One set was MSP for methylated DNA (mDNA): MSP-*HSPB6*-F (GTTTTTAGTAGTTTTTCGTCGAAGC) and MSP-*HSPB6*-R (AAATCCCTATACCTATACAACCGTC). The other set was MSP for unmethylated DNA (UmDNA): UMSP-*HSBP6*-F (TTTTAGTAGTTTTTTGTTGAAGTGT) and UMSP-*HSPB6*-R (AAATCCCTATACCTATACAACCATC). The PCR reaction was prepared as follows: 1 μL bisulfite-treated DNA, 0.5 μL of 10 μM of each primer, and 10 μL DreamTaq Green PCR Master Mix (2X) (Thermo Fisher). The PCR conditions were as follows: (1) one cycle at 95 °C for 5 min; (2) thirty-six cycles at 94 °C for 30 s, followed by 30 s at 57 °C and then 30 s at 72 °C; and (3) one cycle for 10 min at 72 °C. Finally, the PCR products were loaded into a 1.5% agarose gel containing 0.5 g/mL ethidium bromide [36].

### 2.7. RNA Extraction, cDNA Synthesis, and qRT-PCR

Based on the manufacturer’s protocol, QIAZol Lysis reagent (Qiagen) was applied to isolate RNA from 20 colon samples extracted from patients with CC, cultured cells (LoVo and Caco-2), and colon samples of Wistar rats that included the control and DMH-induced groups. Next, the concentration and purity of RNA were measured using the Nano-Drop8000 spectrophotometer, followed by employing GoScript Reverse Transcriptase (Promega) to convert 1 μg RNA to complementary DNA (cDNA) via an RT-PCR system [11]. Gene-specific primers (Table 1) were utilized for qRT-PCR (GoTaq qPCR; Promega) on the Prime Q machine (Techine), normalizing the abundance of the target gene to the housekeeping gene (*GAPDH*). Some assays were normalized to compare the expression of untreated samples [37]. The PCR conditions were 1 cycle at 95 °C for 5 min and then 36 cycles at different temperatures: 95 °C, 58 °C, and 72 °C for 30 s, followed by incubation by 1 cycle of 95 °C for 60 s, 58 °C for 30 s, and 95 °C for 30 s, respectively. The 2−ΔΔCt method was employed to measure the relative mRNA expression fold changes.

### 2.8. Statistical Analysis

SPSS software Ver.22 (SPSS Inc., Chicago, USA) was employed to define the statistical significance. The results were performed in triplicate, and the average ± SD was calculated. The unpaired Student’s *t*-test was statistically significant at ∗ *p* < 0.05; ∗∗ *p* < 0.01.

## 3. Results

### 3.1. Clinical Data of the Enrolled Patients

In our study, the age of the patients ranged from 24–79 years, and the average was 57.8 years. A total of 40% of the patients were younger than 57.8 years old, while the remaining 60% were older than this age (Table 2).

### 3.2. HSPB6 Is Downregulated in CC and Associated with Poor Outcomes

The expression of *HSPB6* in CC relative to NC, as well as the link between *HSPB6* presence and CC patients’ outcomes, was revealed using R2 genomic analysis and a visualization platform (https://hgserver1.amc.nl/cgi-bin/r2/main.cgi, accessed on 2 September 2022). According to this study (GSE8671), which contained a cohort of 32 samples taken from patients with CC and their adjacent NC (Figure 1A), *HSPB6* was significantly downregulated in CC compared to NC. In addition, its high expression was correlated with good outcomes, whereas its low expression was correlated with poor outcomes (Figure 1B). Therefore, these data indicate the effect of *HSBP6* absence on CC aggressiveness.

### 3.3. HSPB6 Depletion Is Observed in Colon Tumorigenesis

The RNA expression of *HSPB6* was examined in CC tissues relative to its adjacent NC tissues as well as in in vivo experiments. The expression of human *HSPB6* was downregulated in CC tissues compared to its adjacent NC tissues, and the reduction in its expression was around 60% (the average *HSBP6* expression in NC tissues was 8.4 ± 2.3, while in CC tissues it was 3.5 ± 1.3) (Figure 2). Previously, we extracted NC treated with saline- and DMH-induced CC from a male Wistar rat [31]. Thus, before we investigated *Hspb6* expression, the proliferation marker *Ccnd1* was examined [38], which was significantly elevated in CC, as well as the differentiation marker *Ndrg2* [39], which was decreased in CC. Our results showed that *Hspb6* was significantly downregulated in DMH-induced CC (Figure 3). Therefore, this finding showed that *HSPB6* expression was downregulated in CC tissues, suggesting that depletion of *HSPB6* exacerbates tumor progression.

### 3.4. 5-aza-2′-Deoxycytidine Induces the Expression of HSPB6

Methylated *HSBP6* [29] and the alternation of the methylation level cause a change in its expression [22]. Thus, we evaluated the methylation status of the *HSBP6* promoter region and its involvement in regulating its expression in LoVo and Caco-2 cell lines that do not express *HSPB6* [40]. Our results indicated that when *HSBP6* was methylated (Figure 4) and LoVo and Caco-2 cell lines were treated with 1 μM of DNA methylation inhibitor (5-aza-2′-deoxycytidine) for 72 h, *HSPB6* expression was significantly elevated (Figure 5). Our results indicated that DNA methylation could contribute to silencing *HSBP6* in CC.

### 3.5. HSPB6 Gene and Protein Interactions

We employed the GeneMANIA database (genemania.org, accessed on 15 October 2022) to uncover the associated genes and proteins that interacted with *HSPB6*. Thus, to construct the data, we selected homo sapiens and chose the automatically selected weight method for our query. The constructed data shows that a total of 10 genes were linked to interact with *HSBP6* at the transcriptional level and were classified according to their interaction with *HSPB6* from the strongest to weakest, as follows: *ITGA7*, *HSPB7*, *DMPK*, *TNS1*, *SORBS1*, *FLNC*, *PPP1R3C*, *FXYD1*, *LMOD1*, and *CASQ2*. In addition, 10 genes were linked to interact with HSPB6 at the transitional level and classified according to their interaction with HSPB6 from the strongest to weakest, as follows: HSPB1, HSPB8, HSD17B10, MAP1B, ATG4A, PNMA2, BAG3, YWHAG, CRYAB, and UBR3 (Figure 6).

## 4. Discussion

In the formation of CC, several genetic and epigenetic alternations have been observed, causing a change from normal cells to cancer cells through the activation of oncogenes and silencing of tumor suppressor genes [41]. Thus, studies have aimed to identify new molecular genetic changes in cacogenesis [41]. Heat shock protein members were identified as aberrantly expressed in CC samples, suggesting their role in colon tumorigenesis, including *HSP70* and *HSP40*, which were upregulated in CC tissues compared to NC [42]. In addition, HSP70 and HSP110 were associated with CC aggressiveness [43]. Moreover, *HSP20* (*HSPB6*) expression has been linked to several cancers [20,21,23,27] and was absent in CC [21]. Epigenetic alternations are also involved in its regulation [22,29]. In this investigation, we found, for the first time, that *HSPB6* expression was downregulated in cancer tissues compared to its adjacent normal tissues and that its expression was decreased in DMH-induced colon tissues in an in vivo study. Furthermore, *HSPB6* was methylated in CC cell lines (LoVo and Caco-2), and its expression was induced after treatment with a DNA methylation inhibitor (5-aza-2′-deoxycytidine).

Our findings indicated that *HSPB6* expression was depleted in CC and in the DMH-induced colon in vivo study [31], suggesting the possible role of *HSPB6* in enhancing tumorigenesis [21,44]. This is in accordance with a report revealed that restoring its expression in HCT116 cells resulted in an induction of the apoptosis pathway [21]. Interestingly, similar data have shown that *HSPB6* expression is downregulated in ovarian cancer, hepatocellular carcinoma, and CC [21,24,28]. In contrast, its overexpression was implicated in non-small cell lung carcinoma tumor growth [45]. We also showed that *HSPB6* was methylated in CC cell lines (Caco-2 and LoVo) and employed a DNA methylation inhibitor [46], leading to upregulated *HSPB6* expression, implying that *HSPB6* was epigenetically regulated by DNA methylation. This is in accordance with [22,29], which showed *HSPB6* expression at the transcriptional level was inversely correlated to the methylation level.

*HSPB6* has a suppressive role in preventing CC tumor growth, and its depletion could stimulate CC progression through several mechanisms. *miR-320* expression has been shown to be elevated in colon diseases [47], which causes a reduction in *HSBP6* [48]. These result in blocking apoptotic events through the increase in the expression of the antiapoptotic proteins Bcl2 and Bcl-xL, causing the inactivation of the expression of caspase3/7 [21]. Furthermore, other reports have demonstrated that HSPB6 inhibits the expression of MAPK, AKT [49], and P13K [50] and interacts with HSP27 to inhibit ERK and hamper hepatocellular cell line progression [51]. Our geneMANIA result indicates the strongest genetic interaction with *HSPB6* is *ITGA7* [52,53], which has been upregulated in cells derived from embryonic stem cells [54]. Interestingly, *HSPB6* [21] and *ITGA7* [55,56] were downregulated in colorectal cancer, suggesting potential transcriptional positive regulation. In contrast, HSPB1 has the strongest physical interaction with HSPB6 [57,58]. We noted that there was an adverse association between HSPB6 and HSPB1, where the expression of HSPB1 was elevated in most cancers [59] and that of HSPB6 was decreased [24]. Additionally, HSPB1 expression is eliminated when HSPB6 is present in different cancer cell lines [40], and the interruption of heterodimer formation between HSPB1 and HSPB6 may lead to disease severity [60]. Notably, the HSPB1 function is mainly regulated by its association, dissociation, oligomerization, and dimerization dynamics [61]. A strong finding has also indicated that HSPB1 and HSPB6 formation results in a reduction in the progression of hepatocellular carcinoma [62,63]. Thus, this association could affect the decision between apoptosis and proliferation [59].

Taken together, we concluded that *HSPB6* expression was downregulated in CC samples compared to adjacent NC as well as in DMH-induced Wistar rats, suggesting its suppressive role. We also noticed that its expression was induced by 5-aza-2′-deoxycytidine treatment. Thus, *HSPB6* could be epigenetically regulated, and its presence may inhibit CC progression by activating the apoptosis pathway [21] by downregulating the expression of HSPB1 at the translation level, which plays a role in inhibiting apoptosis activity by interrupting the BAX and BID protein functions as well as hampering the release of cytochrome c, leading to the deactivation of the caspase cascade [61].

This investigation had some limitations, including the limited number of samples extracted from the patients. Therefore, expanding the number of samples is required for further confirmation of *HSPB6* expression. The size of the samples was also small; consequently, DNA and protein were not extracted. Thus, adequate tissue needs to be provided to study DNA methylation and protein expression.

## Figures and Tables

**Figure 1 medicina-59-00996-f001:**
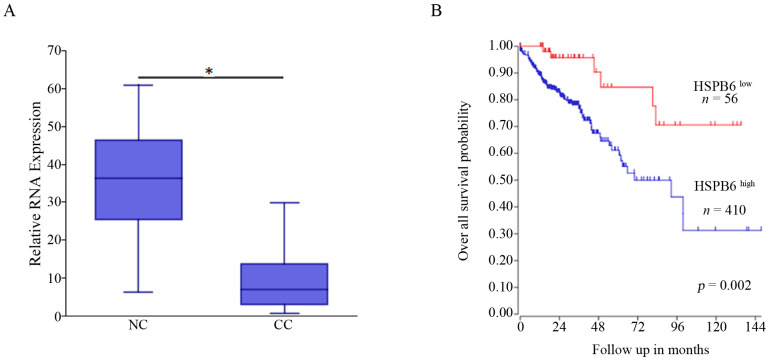
*HSPB6* expression and its association with CC patients’ overall outcomes were investigated using the R2 genomic analysis and visualization platform (https://hgserver1.amc.nl/cgi-bin/r2/main.cgi, accessed on 2 September 2022). (**A**) The box plot indicates 32 patients, including 32 NC and their adjacent CC (GSE8671); low *HSPB6* expression was significantly present in CC relative to its adjacent NC (* *p* = 0.03). (**B**) The Kaplan–Meier survival estimate in 466 samples extracted from microarray analysis of mixed colon adenocarcinoma (GENCODE36) displayed, at cutoff value 1.33 and scan mode, that high *HSPB6* expression was associated with good outcomes and an increase in the overall survival probability. n: number of samples. Abbreviations: NC: Normal Colon; CC: Colon Cancer.

**Figure 2 medicina-59-00996-f002:**
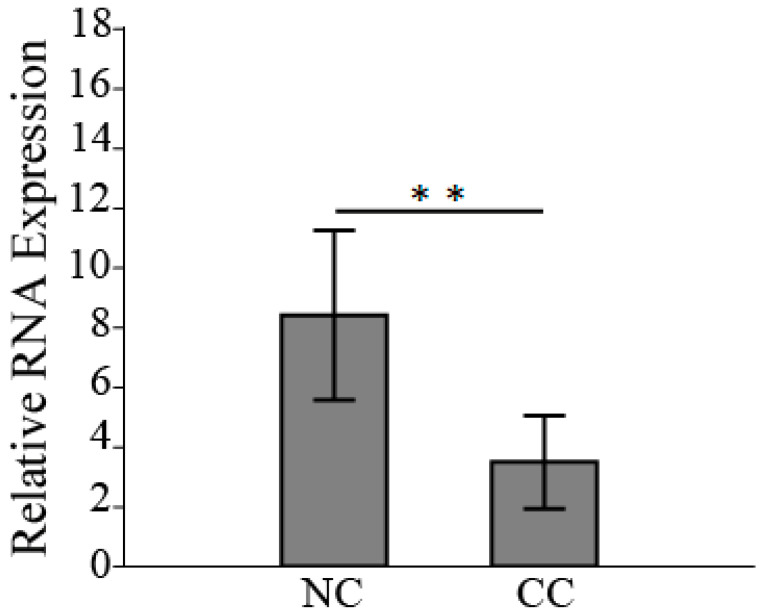
*HSPB6* was poorly expressed in patients with CC. The average expression of *HSPB6* in NC (10 samples) was elevated compared to its adjacent CC (** *p* = 0.002). The mean and SD of the three independent experiments were obtained. The unpaired Student’s *t*-test was applied to define significance.

**Figure 3 medicina-59-00996-f003:**
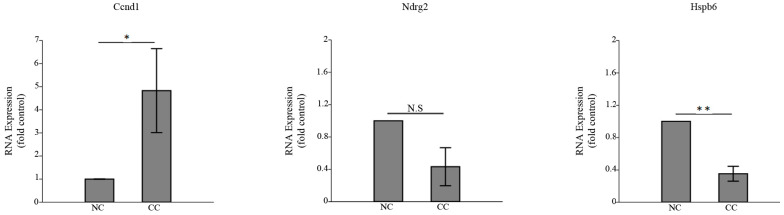
*Hspb6* expression was detectable in tumorigenesis in Wistar rats. The expression of *Ccnd1* (**left panel**), *Ndrg2* (**middle panel**), and *Hspb6* (**right panel**) was examined in NC- and DMH-induced CC isolated from Wistar rats. The expression of Ccnd1 was significantly 4-fold upregulated (* *p* = 0.02), and *Ndrg2* expression was decreased in CC compared to NC (N.S; not significant). *Hspb6* expression was also significantly poorly presented in the CC group compared to the NC group (** *p* = 0.003). The mean and SD of the three independent experiments were obtained. The unpaired Student’s *t*-test was applied to define significance.

**Figure 4 medicina-59-00996-f004:**
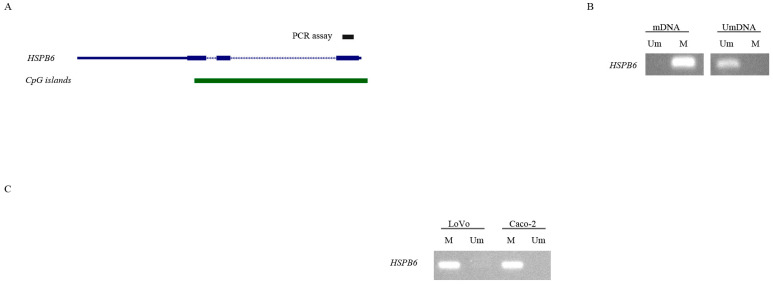
*HSPB6* contains rich CpG islands at the promoter region and is methylated in LoVo and Caco-2 CC cell lines. (**A**) This shows the *HSPB6* gene and the location of the Polymerase Chain Reaction assay, where the amplicon was amplified, and the distribution of CpG islands. (**B**) In the assay controls, the mDNA was fully methylated DNA and the UmDNA was unmethylated DNA. Um and M represent the unmethylated-specific primers and methylated-specific primers, respectively. (**C**) *HSPB6* was methylated in the CC cell lines LoVo and Caco-2.

**Figure 5 medicina-59-00996-f005:**
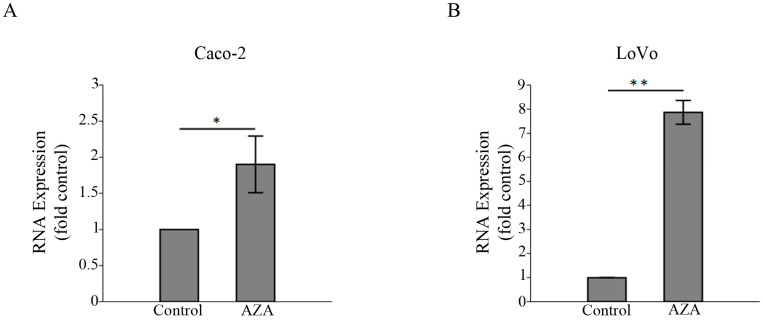
5-aza-2′-Deoxycytidine-stimulated *HSPB6* expression. (**A**) The expression of *HSPB6* was significantly increased in Caco-2 cells treated with 5-aza-2′-deoxycytidine (AZA) compared to Caco-2 cells treated with DMSO (control) (* *p* = 0.02). (**B**) The expression of *HSPB6* was significantly increased in LoVo cells treated with 5-aza-2′-deoxycytidine (AZA) compared to LoVo cells treated with DMSO (** *p* = 0.002). The mean and SD of the three independent experiments were obtained. The unpaired Student’s *t*-test was applied to define significance.

**Figure 6 medicina-59-00996-f006:**
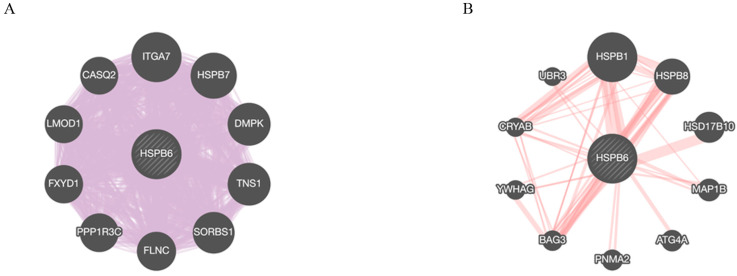
*HSPB6* interaction at transcriptional and translational levels using the GeneMANIA database (genemania.org, accessed on 15 October 2022). The black nodes represent the 10 most important genes. (**A**) The diagram displays the interaction at the transcriptional level between *HSPB6* and other genes, which are ordered from the strongest to weakest interaction, as follows: *ITGA7, HSPB7, DMPK, TNS1, SORBS1, FLNC, PPP1R3C, FXYD1, LMOD1,* and *CASQ2*. (**B**) The diagram displays the interaction at the transitional level between HSPB6 and other genes, which are ordered from the strongest to weakest interaction, as follows: HSPB1, HSPB8, HSD17B10, MAP1B, ATG4A, PNMA2, BAG3, YWHAG, CRYAB, and UBR3.

**Table 1 medicina-59-00996-t001:** Primer sequence used for qRT-PCR.

Gene	Sense Region (5′–3′)	Antisense Region (5′–3′)	Ta ^1^	Product Size (bp)
*HSPB6*	CTTCCAGACCCTACCAGCAC	CCAAGAAAGTGGGTCGGTGG	58	71
*GAPDH*	GGGAAGCTTGTCATCAATGG	GAGATGATGACCCTTTTGGC		173
*Ccdn1*	TCCGGAGACCGGTCGTAGA	GTCTTAAGCATGGCTCGCAG		194
*Ndrg2*	GCTTGTTGTCCAACTTCTCAC	GAATGAGTCTGTCCCTGGTCC		192
*Hspb6*	ATGAGGAGCGCCCAGATGAA	GGAGACAGTGCAGAGGTCAC		106
*Gapdh*	CAACTCCCTCAAGATTGTCAGC	GGCATGGACTGTGGTCATGA		180

^1^ Annealing temperature.

**Table 2 medicina-59-00996-t002:** General characterization and clinical data of the enrollment patients.

Parameters		NC		CC
Number of patients		10 (100%)		10 (100%)
Average of the age (youngest–oldest)		57 years and 8 months (24–79)		
Above 57 years and 8 months		6 (60%)		6 (60%)
Below 57 years and 8 months		4 (40%)		4 (40%)
The age, gender, TNM, and cancer grade of each patient				
Number of patients	Age	Gender	TNM staging	Cancer Grade
1	79	Male	T2N1M0	2
2	54	Male	T3N0Mx	2
3	24	Male	T3N0Mx	2
4	69	Male	T2N2M0	2
5	61	Male	T2N0Mx	2
6	38	Male	T3N1Mx	1
7	73	Male	T2N0Mx	2
8	79	Male	T3N0Mx	2
9	38	Male	T3N1Mx	2
10	63	Male	T3N0Mx	2

Abbreviations: NC: Normal Colon; CC: Colon Cancer; TNM: Tumour, Node and Metastasis.

## Data Availability

The original contributions presented in the study are included in the article; further inquiries can be directed to the corresponding author.

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
