# Peer review of "HSPB6 Is Depleted in Colon Cancer Patients and Its Expression Is Induced by 5-aza-2′-Deoxycytidine In Vitro"

_medicina, 2023, doi:10.3390/medicina59050996_

Round 1

Reviewer 1 Report

Authors propose HSPB6 as a new biomarker for colorectal cancer (CC). They have obtained very interesting information although it requires some  clarifications.

1) Authors did not describe what they consider NC tissue. At what minimum distance from the tumor region has the normal sample been collected from the tumor sample? Please, add some of this information in Materials and Methods, section 2.2.

2) In table 2, authors showed some of the characteristics of CC patients. I encourage authors to make a more detailed table with each patient (as they are just 10 patients), showing at least age, gender and American Joint Committee on Cancer (AJCC) stage of each patient.

3) Throughout the text, all the names of genes as HSPB6, Ccnd1...should be written in italics. Please, could the authors change it?

4) In Figure 1B, authors showed a Kaplan-Meier where the patients are separated in two groups: low or high HSPB6. However, they did not explain how these two groups where made. What do they consider low or high? Where did they put the cutoff point? I encourage authors, to add a detailed information in a new section of Materials and Methods.

5) Could the authors also add the name of the database where they got the 466 patients of figure 1B?

No comments

Reviewer 2 Report

In their manuscript Almutairi et al. have evaluated the level of HSPB6 mRNA in colon cancer and in adjacent normal tissues collected from patients suffering from colon cancer as well as in DMH-induced Wistar rats. Additionally, the authors demonstrated that the methylation status of the HSBP6 promoter region is closely associated with its expression in LoVo and Caco-2 cell lines.

However, there are several comments for the authors:

1) In my opinion, the problem is the small study group.

2)      Title of study “HSPB6 is Depleted in Patients with Colon Cancer Among Saudi Population and Its expression is induced by 5-aza-2’-deoxycytidine” is confusing and should be corrected. The study group is too small to apply conclusions to the entire Saudi population. Furthermore the effect of 5-aza-2’-deoxycytidine on HSPB6 expression has been evaluated in colon cancer cells but in in vitro study.

3)      In the Materials and methods section, the Authors should report the number of cells seeded in in vitro experiment.

4)      Did the authors assess the quality and quantity of nucleic acids extracts? If relevant, what method was used? The information should be added.

5)      In the “Clinical data of the enrolled patients” section should be added more information about study group, e.g. histological grade and clinical stage of tumor.

6)      Are the results in the Figs 1A and 2A presented as average ± SD? Do box plots show median, quartiles, and minimum and maximum values?

7)      Both graphs 2A and 2B show significantly downregulated expression of HSPB6 mRNA in CC (10 samples) in comparison to their adjacent NC, so Figure 2A should be deleted.

8)      The HSPB6 interactions between other genes and proteins should be more described in “HSPB6 gene and protein interactions” chapter and in the Discussion section.

9)      The methylation status of the HSPB6 promoter region should be detected in patient samples too.

Round 2

Reviewer 2 Report

The authors have addressed convincingly the points that were raised in the review. It can be accepted in this form.